# Income Inequality, Institutions, and Freedom of the Press: Potential Mechanisms and Evidence

**Umut Uzar** 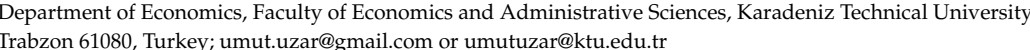

Department of Economics, Faculty of Economics and Administrative Sciences, Karadeniz Technical University, Trabzon 61080, Turkey; umut.uzar@gmail.com or umutuzar@ktu.edu.tr

**Abstract:** In the last few decades, income distribution has deteriorated in a large part of the world. The inability to stop inequality has evolved into a major social crisis and has become one of the most urgent issues globally. Given the importance of the issue, identifying the root causes of inequality can be a guide for policy makers in solving the problem. Although there are a few studies linking institutional quality with income inequality in recent years, the question of whether freedom of the press affects income distribution remains unanswered. This study is the first attempt to address this question. With this motivation, the study researches the influences of institutional quality and freedom of the press on income inequality for the BRICS-T (Brazil, Russia, India, China, South Africa, and Turkey) countries for the period 1993–2016. Moreover, globalization, economic growth, and trade openness are included in the model to avoid the problem of omitted variable bias in explaining inequality. The study findings indicate that institutional quality and freedom of press, which are the main independent variables for the entire panel, reduce inequality. In addition, although trade openness is a factor that reduces inequality, globalization and economic growth are not statistically significant. Although the country-specific estimates show heterogeneity, they are quite promising in terms of inequality, institutional quality, and freedom of the press. In this framework, policy makers can reduce inequalities by designing policies that emphasize institutional quality and freedom of the press. With such a win-win opportunity, BRICS-T countries can achieve two important gains to reach developed country status.

**Keywords:** income inequality; institutional quality; freedom of press; globalization; sustainable development

## 1. Introduction

Although income inequality between countries has decreased in the last few decades, this is not the case for income inequality within countries. The degree of income inequality has increased significantly in many countries, irrespective of their level of development, and it has emerged as a pressing global concern [1,2]. Current evidence indicates that despite impressive economic performance of the world economy during the last several decades, the benefits of growth have not been distributed fairly [3,4]. In fact, the disappearance of dynamics that ensure fairness in income distribution since the 1980s has pushed income inequality beyond its historical level in developed and developing countries, despite high growth rates [5]. Income and wealth inequalities in developed countries, particularly in the United States and OECD countries, have been quite remarkable in recent decades. According to Saez and Zucman [6], enhancing income inequality has increased the top 0.1% share of total wealth in the United States from 7% to 20% in the last 30 years. Similarly, income inequality has also increased in many developing countries during this period [7,8].

As mentioned, the inability to trickle down the fruits of economic growth has led to the simultaneous realization of growth and inequality. Furceri and Ostry [5] emphasize that moderate inequality in market economies can have a positive influence on investment and growth, but a high increase in the level of inequality can have destructive effects on

economic growth and sustainability. In this context, high inequality is deeply dangerous in terms of investments, economic growth, and poverty [9]. Moreover, increasing inequality can lead to asymmetric distribution of power among social groups and result in some negative consequences. Within this context, asymmetric effects promote populist policies, disrupt macroeconomic stability, increase financial fragility, and trigger crises [10,11]. In addition to its economic implications, its negative impact on political stability and social cohesion makes increasing income inequality one of the most significant obstacles to sustainable development.

Reducing inequality is not only substantial for promoting a fairer distribution of income and strengthening social cohesion but also highly strategic for supporting sustainable growth [5]. Therefore, the size of inequality, its root causes, and potential solutions have become one of the most debated topics globally by both researchers and policymakers. In fact, among heterodox economists, issues such as income generation and distribution, which are at the center of political economy, have been discussed theoretically and empirically for a long time, while among mainstream economists, income inequality has been pushed into the background due to political/ideological concerns. However, the increasing problem of income inequality in a significant part of the world and the rising economic and social costs of this deterioration have led to income inequality becoming one of the most crucial areas of discussion in mainstream literature as well [12]. The fact that Thomas Piketty's [3] Capital in the Twenty-First Century and Joseph E. Stiglitz's [12] The Price of Inequality: How Today's Divided Society Endangers Our Future have been on the bestseller lists around the world for a long time can be seen as concrete evidence of public interest in inequality.

Articles on the main determinants and outcomes of the income distribution, and especially income distribution policies, are increasing rapidly in the international literature. Early studies investigating the root causes of income inequality often focused on economic growth/economic development. Kuznets's [13] seminal paper focused on the connection between economic growth/development and income inequality. Over time, the number of studies explaining income inequality has increased rapidly and the explanatory variables have diversified. Globalization [14], monetary and fiscal policies [15], macroeconomic framework [16], financial development [17], technological change [18], business globalization [19], financial globalization [20], and education [21] have been widely used to explain inequalities.

Recently, with the discovery of the important influences of institutional quality on the economic structure, institutions have begun to be integrated into inequality studies at an increasing rate. In this framework, it has begun to be investigated whether institutions can be a key factor in reducing income inequality [5,9,22–27]. These initiatives have provided evidence that institutional quality indicators such as government stability, corruption control, law, bureaucratic quality, and democracy can have an impact on inequalities by preventing market failures and operating redistribution mechanisms. Although these studies highlighting institutional factors have provided unique insights into explaining income inequality, there are still some limitations. Studies have generally concentrated a small part of institutional quality such as corruption control and government effectiveness [28]. This approach may pose a risk of not fully capturing the diversity of institutions and their impact on income distribution.

While new initiatives have emerged that prioritize investigating the institutional roots of income inequality, the number of studies that examine the drivers of income inequality remains limited and tends to concentrate on established variables [5]. As mentioned, previous studies have generally focused on a range of economic, political, and social factors. This has led to a saturation point in studies that examine the determinants of income distribution. When looking at previous literature, press freedom, which is considered an important indicator of democratization and could have potential effects on inequality [29], has not been considered as a determinant of income distribution and has been neglected by researchers. Press freedom is one of the most substantial parts of civil society and, after the legislature, the executive, and the judiciary, is the fourth power in democracies. With

these aspects, press freedom can act as a catalyst for solving social issues and achieving sustainable development goals [30]. In fact, this situation raises an important question for researchers: can press freedom really reduce income inequality? Since it has not been addressed in previous literature, this study assigns a specific role to press freedom in explaining income inequality. In light of these developments, the aim of this study is to examine the impacts of institutional quality and press freedom on income distribution in BRICS-T countries during the period 1993–2016. Additionally, globalization, economic growth, and trade openness, which have been used in the literature to explain inequality, are included in the model to avoid the problem of neglected variables.

The objective of this study is to make several noteworthy contributions. First, there is no consensus on the impact of institutional quality on inequality. This study creates an inclusive institutional quality variable by forming an index from 6 indicators obtained from the International Country Risk Guide (ICRG). Through this index, it reveals more comprehensively the impact of institutions on income inequality. Second, the selection of BRICS-T countries is based on some unique justifications. These countries have achieved higher growth rates compared to developed countries such as OECD, yet income inequality has risen [27]. In addition, it can be said that the institutional framework and freedom of the press are still not at universal standards in these countries [30]. All these developments make the BRICS-T countries a unique laboratory to research the connections between inequality, institutions, and press freedom. Thus, the fresh evidence to be obtained from this study can be a strong guide for policymakers in the fight against inequality. Third, the relationship between press freedom and income inequality has not been addressed at the theoretical and empirical levels in the literature before. To the best of our knowledge, this study is the first of its kind and serves as a catalyst for the establishment of a new research network. If evidence is obtained that press freedom reduces income inequality, policymakers will have the opportunity to expand freedoms and reduce inequality.

In the remaining part of the study, potential mechanisms between the main variables, literature, dataset/methodology, findings, discussions, and conclusions/policy recommendations are presented.

## 2. Income Inequality, Institutions, and Freedom of Press: Potential Mechanisms

Although there are many factors that determine income distribution, it is suggested that well-functioning institutions can perform a substantial role in curbing income inequality [25]. Chong and Gradstein [24] state that countries with weak institutions are likely to have high levels of inequality along with macroeconomic problems. However, despite the potential mechanisms proposed by economics theory to explain inequality, there is little agreement among them [5]. In fact, the relationship between institutions and income distribution is quite complex, and there is no consensus on potential mechanisms [26].

According to North [31], institutions can be broadly defined as the fundamental principles governing social interactions. Within this framework, institutions establish the fundamental rules that limit and guide the behavior of economic actors. These institutions can be formal, such as constitutions, laws, and regulations, or informal, such as norms and traditions. Examples of these institutions include corruption control, the rule of law, government effectiveness, accountability, bureaucratic quality, traditions, and customs. Establishing a strong institutional infrastructure is a key factor for long-term economic goals such as economic growth and development, as well as social cohesion [32–34]. Institutions may have the ability to reduce income inequality based on their capacity to prevent market failures and activate redistribution mechanisms [26]. Thus, institutions can influence both the primary income distribution generated by the market and the secondary income distribution (post taxes and transfers). On the other hand, political instability, corruption, nepotism, and inadequate property rights indicate low institutional quality. Therefore, in such countries, the interests of certain groups are prioritized over those of society, and income inequality deepens.

The first potential mechanism linking institutions and income inequality is property rights. Property rights are associated with the rule of law and guaranteed by it [35]. Moreover, the rule of law can reduce inequality by curbing opportunity inequality and nepotism [36]. Property rights guaranteed by law encourage foreign direct investment, domestic investment, and portfolio investment. Increased investments in different categories are the engine of economic growth. Milanovic [37] notes in his seminal book that economic growth is one of the most substantial tools for reducing global and national inequality. Particularly in poor countries, economic growth plays a crucial role in improving individuals' living standards. This is because economic growth enables new investments to be made, employment to increase, and wages to rise. More importantly, economic growth increases the government's tax revenues and its capacity to make social expenditures. In this context, the financial resources created by growth facilitate the implementation of active redistribution policies. Therefore, an improvement in property rights can be a dynamic that revitalizes economic activity and curbs income inequality [38].

Another category that can be associated with the connection between institutions and income inequality is corruption control. In nations where corruption is widespread, the elites with economic and political power can use resources for their own interests through corrupt politicians and bureaucrats. The inefficient allocation of resources hinders economic growth and prevents income from trickling down. Furthermore, the elites can facilitate the construction of a biased and self-serving tax system through corrupt politicians, which makes it difficult to build a progressive tax system that is an important tool for reducing inequality [39]. Distortions in the tax system and reduced tax revenues lead to reduced spending in areas such as education, health, and social services, which are significant pillars of income inequality [28].

Institutional indicators such as government stability, democracy, accountability, and bureaucratic quality have the potential to affect income distribution. Improvements in these categories increase the efficiency of government policymaking [40]. Effective and efficient policies can facilitate the design of better redistribution policies. In this context, institutions of this kind can implement more efficient policies on issues such as redistribution, poverty reduction, and limiting the privileges of the elite [5]. For example, weaknesses in such institutional indicators may lead to economic policies being designed in favor of the elites by increasing lobbying activities [41]. Policies that prevent such lobbying activities enable political power to spread to wider sections of society. Acemoglu et al. [42] state that eliminating power asymmetry creates an equalizing effect. This is because the spread of political power to wider sections of society through institutions can strengthen the trend towards redistribution and poverty-centered policies, thereby reducing inequality.

Although there is no complete consensus on the potential influences of institutions on income distribution, as mentioned above, many researchers have tried to designate potential mechanisms. However, the effects of press freedom are quite mysterious. Therefore, the potential mechanisms between press freedom and income distribution are a completely missing link. Nevertheless, press freedom can perform a strategic role in the interests and well-being of society in many ways. Press freedom is an integral part and guarantee of democracy, institutions, and civil rights [40,43]. In this respect, it is an important parameter in controlling corruption, expanding political participation, and ensuring accountability [44]. Additionally, press freedom acts as an important bridge between ordinary citizens and the government. It captures, criticizes, creates pressure, and corrects a realistic picture of all kinds of activities in society. The roles that the press undertakes can reduce uncertainties and power asymmetries by creating a more transparent and democratic society, and the elimination of many issues, including inequality [45].

In fact, the most critical function of press freedom is to prevent the accumulation of power in a certain group and to limit that power [40]. In other words, it is a catalyst that reduces the power asymmetry between different social groups. In this context, a free press can direct institutions to distribute power equally through public pressure. Thus, it can prevent the political and economic exploitation of the poor by the rich elite. A free press

exposes activities that are not beneficial to society and creates pressure for their elimination. For example, when income inequality worsens due to insufficient redistribution policies, there is a need for a pressure element that will revise and improve redistribution policies. If a free press presents the issue of inequality and its causes to society well, it can constitute serious public pressure. In this context, increased public pressure can lead the government to review and correct its policies. As seen, press freedom can improve the decision-making process of the government by shaping public opinion. Restricting actions that are contrary to the interests of the middle class and the poor is important for reducing inequalities. In this context, the press can play an important role in implementing reforms that eliminate the privileges of some groups in society. The conjuncture resulting from the reform process can be beneficial for both primary and secondary income distribution. Therefore, press freedom can contribute to reducing income inequality by playing a substantial role in the realization of revolutionary reform movements.

Redistribution is almost entirely functional in ensuring justice in income distribution. Additionally, the nature of redistribution is largely dependent on the power relations between different societal groups. Acemoglu et al. [46] and Josifidis et al. [25] note the negative effects of limiting democracy on redistribution policies and income distribution. In this context, the development of democratic values such as institutional quality and press freedom plays a dominant role in spreading power to the people. A balanced distribution of power can address market failures and facilitate better redistribution policies. Therefore, freedom of the press has the potential to affect inequality as an actor that disseminates power to society.

Finally, Figure 1 summarizes the potential channels of institutions and press freedom on income distribution. Concordantly, the institutional framework created by institutions and press freedom affects income distribution through property rights, equal opportunities, equal distribution of power to society, and effective economic policies. Positive developments in these categories create an economic, social, and political environment that can enable a fairer distribution of power and income. As a contribution to Figure 1, the potential mechanisms/functions and tools of press freedom on income equality are provided in Table A1 in Appendix A.

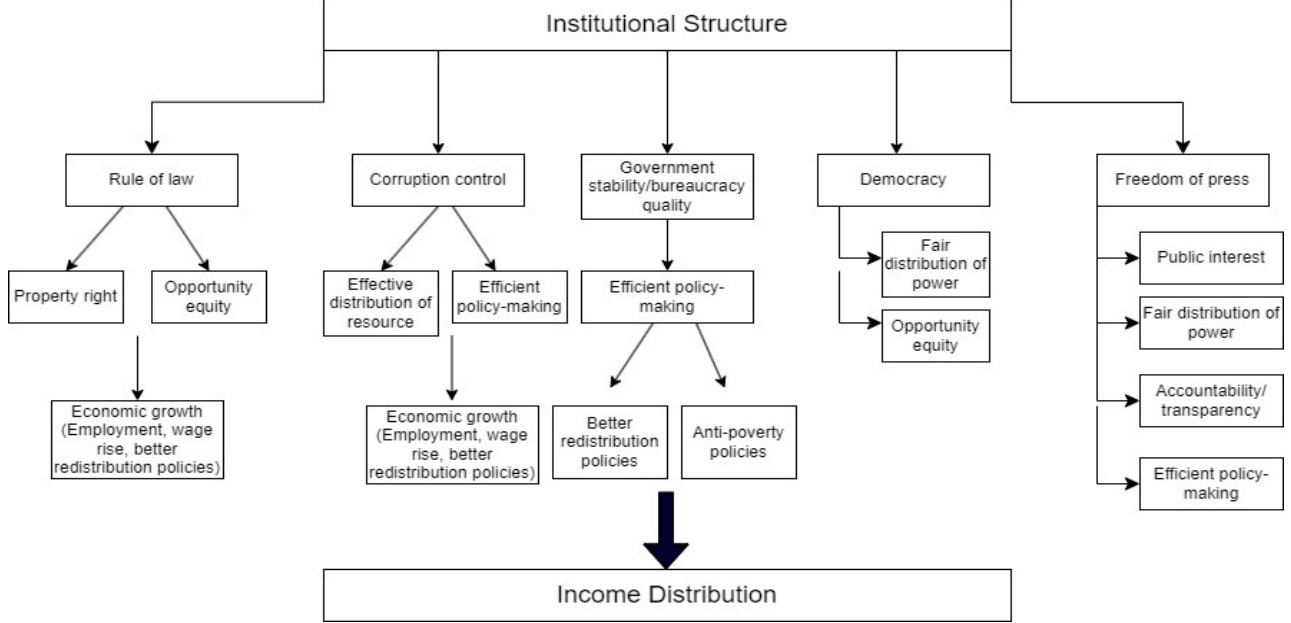

**Figure 1.** Institutional quality, freedom of press, and income inequality: potential mechanisms. Source: Created by the author based on the literature.

### 3. Literature Review

There is a significant effort by policymakers and researchers to explain the reasons for income inequality. In particular, the relationship between fair distribution and institutional quality has become a well-liked research area, especially with the increasing prominence of sustainable development goals in recent years. As a result, institutions have begun to be rapidly integrated into inequality studies. Within this framework, attempts have been made to determine the impact of institutional quality indicators such as democracy, corruption, property rights, law, political stability, bureaucratic quality, and government effectiveness on income distribution. This study focuses basically on institutional quality and press freedom as explanatory factors for income inequality. In this section, studies examining the relationship between institutional factors and inequality are introduced, and the explanatory power of institutions for inequality is examined. Otherwise, as it can be understood from the fact that this study is the first to examine the relationship between freedom of the press and income distribution, there is no direct/indirect theoretical or empirical study in previous literature on this subject.

Studies examining the connection between institutional quality and inequality have placed special emphasis on corruption. A large portion of the literature has examined the impact of corruption on inequality [28]. The literature states that the impact of corruption on inequality is heterogeneous. Some studies have found that corruption increases inequality. Gupta et al. [39] examined the relationship between corruption, inequality, and poverty for a wide set of countries during the period 1980–1997. The study found that the increase in corruption enhances inequality. Gyimah-Brempong [47] focused on the effects of corruption on growth and inequality in 21 African countries from 1993 to 1999. The study showed that corruption is a factor that increases income inequality. Batabyal and Chowdhury [48] investigated the relationship between corruption, finance, and inequality in 30 countries during the period 1995–2008. The study demonstrated that reducing corruption and achieving financial development together would lead to a reduction in inequality. Dincer and Gunalp [49] discovered strong evidence that corruption increased income inequality in the 48 US states during the 1981–1997 period. Some studies emphasize that corruption control alone is not sufficient. For example, Saha et al. [50] provided evidence that corruption control reduces income inequality in 21 Asian countries if education becomes widespread.

Otherwise, there are studies that have found that the increase in corruption through certain transmission channels reduces inequality. Dobson and Ramlogan-Dobson [51] showed that low corruption caused higher income inequality in 19 Latin American countries during the period of 1984–2003. The negative connection between corruption and inequality was explained by the size of the informal economy. Policardo and Carrera [52] studied the causal connection between corruption and inequality in a panel dataset covering 50 countries and the period of 1995–2015. The results demonstrated that the direction of causality could be country-specific and that corruption was not a significant factor in explaining inequality. Keneck-Massil et al. [53] researched the impact of corruption on inequality in countries with different income levels during the period from 1975 to 2017. In the study, a negative relationship was found between corruption and income inequality in developing countries. Furthermore, Malla and Pathranarakul [54] concluded that corruption and government effectiveness do not affect income distribution in developed and developing countries.

Another factor that is given special importance in research on institution inequality is democracy. Rodrik [55] examined the influence of democracy on manufacturing wages in his groundbreaking study. Within this framework, it was concluded that the level of democratization positively affects wages and reduces inequality. Milanovic et al. [56] investigated the effect of democracy on inequality in 126 different countries with varying levels of development. It was concluded that democratization reduces inequality through redistribution in societies where inequality is not highly emphasized. Albertus and Menaldo [57] analyzed the relationship between democracy and redistribution using a global panel dataset. The findings suggested that democracy operates better as a redistribution mecha-

nism. Anyanwu et al. [9] studied the determinants of income inequality in 17 West African countries from 1970 to 2011, including democracy as a variable added to the model to represent institutions. The study found that democracy had a corrective effect on income distribution in the examined countries. Hassan et al. [58] researched the influence of democratic accountability on income inequality and poverty in Pakistan. The outputs exhibited that democratic accountability reduces both income inequality and poverty.

In addition to studies demonstrating that democratization can reduce inequality in line with expectations, there are also studies indicating that democracy does not always have the ability to reduce inequality. Acemoglu et al. [46] researched the connection between democracy, redistribution, and inequality for a comprehensive dataset. The study findings showed that democracy cannot reduce inequality when it is captured by the elites. It is stated that democracy can reduce inequality when it is not controlled by a particular group. Wong [59] reached a similar conclusion to Acemoglu et al. [46]. In the study, which examined 78 countries, it was revealed that democracy can reduce inequality if certain conditions are met.

In recent years, researchers have begun to focus on different and diverse indicators such as government stability, law, bureaucratic quality, accountability, and transparency in addition to corruption and democracy. Chong and Gradstein [24], one of the early studies in this category, examined the causality relationship between an institutional quality index and income distribution for a global panel set covering the period 1960–2000. The study outcomes indicated a bidirectional causality relationship between the variables. Josifidis et al. [25] investigated the connection between income distribution and institutional quality in 21 OECD countries. The study indicated that inadequate redistribution and inequality were the results of institutional factors. Kouadio and Gakpa [27] analyzed the interaction between economic growth, institutional quality, poverty, and inequality in West Africa for the period 1984–2015. The findings showed that institutional factors were crucial in reducing poverty and achieving a fair income distribution in West Africa in the long term. Szczepaniak et al. [26] analyzed the impact of institutional indicators such as freedom, corruption, government effectiveness, regulation quality, and accountability on inequality in Indonesia. The study showed that each institutional indicator was an important factor in reducing inequality from 1999–2019. Blancheton and Chhorn [60] investigated the influences of public expenditure and institutional quality on income inequality in the Asia Pacific Region in 1988–2014. The outputs designated that institutional quality reduces inequality, and there is unidirectional causality from institutional quality to inequality. In addition, Ullah et al. [61] designated that institutional quality is an important moderation task in reducing income inequality and poverty in 64 One Belt One Road countries.

While these studies suggest that institutions will be an important catalyst in reducing inequality, there is also evidence that institutions cannot create strong effects in reducing inequality. Kunawotor et al. [28] researched the institutional roots of income distribution in Africa from 1990 to 2017. The findings indicate that institutional variables such as government effectiveness, regulatory quality, and political stability did not affect income inequality. Asamoah [4] studied the impact of institutional quality on income inequality in developing and developed countries for the period 1995–2017. According to the study findings, the effects of institutions in reducing inequality are quite uncertain and complex. Additionally, the threshold value of institutions that reduce inequality in developing countries is higher than in developed countries. Batuo et al. [16] researched the drivers of inequality in 52 African countries for the period 1980–2017. Similar to Kunawotor et al. [28], this study found that institutional quality has a limited role in reducing inequalities.

As seen, most of the studies examining the institutional roots of inequality have focused on specific few indicators. Although recent studies have used more comprehensive institutional indicators, the evidence is quite heterogeneous and insufficient in number. On the other hand, there is no study that relates press freedom and inequality. In this sense, the study aims to ensure fresh evidence by examining the impact of institutional quality

and press freedom on inequality in BRICS-T countries and to be the first study on press freedom specifically.

## 4. Model and Data Definitions

This study primarily aims to investigate whether institutional quality and freedom of press have an impact on income inequality in BRICS-T countries. Following previous studies examining the drivers of inequality, globalization [5], economic growth [26,28], and trade openness [4,16] are included in the model. Thus, the aim is to provide a more comprehensive explanation of the fundamental causes of inequality in these countries. Due to data constraints in some countries, the analysis period was determined as 1993–2016. The study employed yearly data and examined all variables in their logarithmic form. In this context, the model showing the relationship between the variables is designed as follows:

$$\ln GINI_{it} = \beta_0 + \beta_1 \ln IQ_{it} + \beta_2 \ln FP_{it} + \beta_3 \ln GLB_{it} + \beta_4 \ln EG_{it} + \beta_5 \ln T_{it} + \varepsilon_{it} \qquad (1)$$

In the model, $\beta_0$ presents constant term; $\beta_1$, $\beta_2$, $\beta_3$, $\beta_4$, and $\beta_5$ reflect coefficients of independent variables; and finally $\varepsilon_{it}$ is error term. GINI is the dependent variable and represents income inequality. The Gini Index created by Solt [62] was obtained from The Standardized World Income Inequality Database. This dataset is a reputable resource that researchers have used in recent years [63]. The Gini index takes values between 0 and 100, where 0 denotes complete equality, while 100 denotes complete inequality. In addition, the inequality data used in the study is the Gini coefficient calculated according to the disposable income. In other words, secondary income distribution (post tax/transfer) data are used. This was chosen because both institutions and freedom of the press have the potential to not only eliminate market failures but also improve redistribution policies. For this reason, it is thought that the effect of institutions and freedom of the press on income distribution will be more clearly demonstrated over the Gini calculated according to disposable income.

IQ represents institutional quality and is one of the main independent variables. These data were obtained from the ICRG. Here, an attempt has been made to create a comprehensive index by following Uzar [40]. Within this framework, "(1) government stability, (2) bureaucracy quality, and (3) corruption control" represent the effectiveness of policymaking, while "(4) military in politics and (5) democratic accountability" represent democratization, and "(6) law and order" represents the supremacy of law and the institutional quality index consists of these 6 indicators. All of these six indicators used are rescaled from 0–10. The index is calculated by adding up these indicators and can range from 0 to 60, with higher values indicating stronger institutional quality. FP stands for freedom of the press, which is the other main independent variable. FP is an index that takes values between 0 and 100. Lower values indicate a freer press, while higher values indicate a non-free press. FP was gathered from Freedom House [64].

In the study, GLB, EG, and T represent control variables. GLB reflects globalization. This comprehensive dataset, obtained from the KOF Swiss Economic Institute through the methodological contributions of Gygli et al. [65], represents the economic, social, and political components of the globalization process. Per capita GDP (in constant 2010 USD) is defined as EG, while trade openness, which represents the proportion of total exports and imports to GDP, is denoted as T. Both variables are obtained from the World Development Indicators (WDI). Moreover, detailed explanations about the calculation method and sources of the variables are provided in Table A2 in Appendix B.

Table 1 shows the descriptive statistics for the non-logarithmic forms of the variables. The lowest and peak values of GINI are 33.3 (Russia) and 63.5 (South Africa) and the mean value is 45.76. The lowest and peak values for IQ are 25.90 (Russia) and 45.97 (South Africa), respectively. The mean of the IQ is 35.06 for the entire panel. Finally, the lowest and peak values of FP are 23.0 (South Africa) and 89.0 (China) and the mean value is 52.8.

**Table 1.** Descriptive statistics.

|       | Mean    | Minimum | Maximum   | Std. Dev. |
|-------|---------|---------|-----------|-----------|
| GINI  | 45.76   | 33.30   | 63.50     | 8.73      |
| IQ    | 35.06   | 25.90   | 45.97     | 4.58      |
| FP    | 52.84   | 23.00   | 89.00     | 20.20     |
| GLB   | 59.22   | 34.79   | 72.03     | 8.72      |
| EG    | 6438.52 | 611.11  | 13,853.10 | 3861.90   |
| T     | 43.52   | 15.63   | 72.86     | 13.77     |

*Methodology*

In this study, the impacts of institutional quality, freedom of press, and control variables on inequality in BRICS-T countries are examined. To obtain reliable estimates in this regard, some econometric procedures will be followed. The first step towards reliable econometric procedures is the examination of cross-sectional dependence (CSD) [66]. It is crucial to determine whether the series have CSD or not. If the CDS analysis is not conducted at the beginning of the econometric analysis, the results may be biased. For instance, if there is CSD and unit root tests are applied without taking this into account, the results will be inaccurate. To ensure reliable results, four commonly used tests in the literature will be performed in this study. These tests include the Breush and Pagan [67] LM test, the Pesaran [66] CD and CDLM tests, and finally the Pesaran et al. [68] LMadj test.

After obtaining the CSD results, the second procedure applied is the unit root analysis. If the series exhibits CSD, the unit root test applied should consider its presence. Otherwise, the stationary levels of the variables cannot be accurately determined by applying unit root tests that do not consider CSD. This can create a chain effect and lead to unreliable results in the later stages of the econometric procedure. Therefore, the Pesaran [69] CADF unit root test, which takes into account CSD, is employed.

The third stage of the econometric procedure is to determine whether there is a cointegration among variables. The existence of cointegration indicates a long-term relationship among variables. The identification of a long-term relationship allows policymakers to design income distribution policies using the relevant variables. The study preferred the Durbin–Hausman (DH) cointegration test developed by Westerlund [70] due to its advantages as a second-generation cointegration test. One of the main advantages of DH is that it accounts for CSD and avoids producing biased results in its presence. Additionally, while the dependent variable should be first-order stationary (I(1)) for the test to be applied, no such requirement exists for the independent variables, which only need to be I(0) or I(1) [70]. Lastly, DH is a two-dimensional test, which enables the homogeneity or heterogeneity of autoregressive parameters to be assessed.

In the concluding phase of the econometric procedure, the AMG (augmented mean group) method proposed by Eberhardt and Bond [71] and Eberhardt and Teal [72] was employed to estimate the long-term coefficients of institutional quality, press freedom, and control variables on inequality in this study. The AMG estimator is widely used in the literature due to some advantages it provides in obtaining long-term coefficients [40]. Firstly, AMG is a second-generation estimator that takes into account CSD. In cases where CSD is present between the series, it provides strong and reliable estimates [73]. The unit root structures of the variables are not important in the application of this method. In other words, whether the variables are I(0) or I(1) does not pose a constraint for the application of this method. Furthermore, according to Hussain et al. [74], AMG is a long-term co-integration estimator that has been enhanced for datasets with a low number of cross-section and periods, providing reliable results. In this regard, the fact that the number of cross-section and periods is not too high in the study justifies the use of the AMG estimator to a significant extent. Lastly, AMG estimates not only the long-term coefficients for the entire panel but also for each country that makes up the panel. This allows for specific policy recommendations to be made for each country.

Figure 2 illustrates the econometric technique utilized, making it simpler for the readers to comprehend the methodological steps. Initially, four separate tests are conducted to scrutinize the existence of CDS. When CDS is present, first-generation econometric techniques fail to deliver dependable outcomes, necessitating the use of second-generation methods. Within the context of CDS existence, Pesaran [69]'s CADF test, a second-generation method, is utilized. If all variables are I(1), an ECM test comes into play. If there is a mix of I(0) and I(1), the Durbin–Hausman test by Westerlund [70] provides a higher level of reliability [75]. In the event of cointegration, one can proceed with long-term coefficient estimation. It should be noted that AMG, being a second-generation estimator, offers substantial benefits in identifying long-term associations [40].

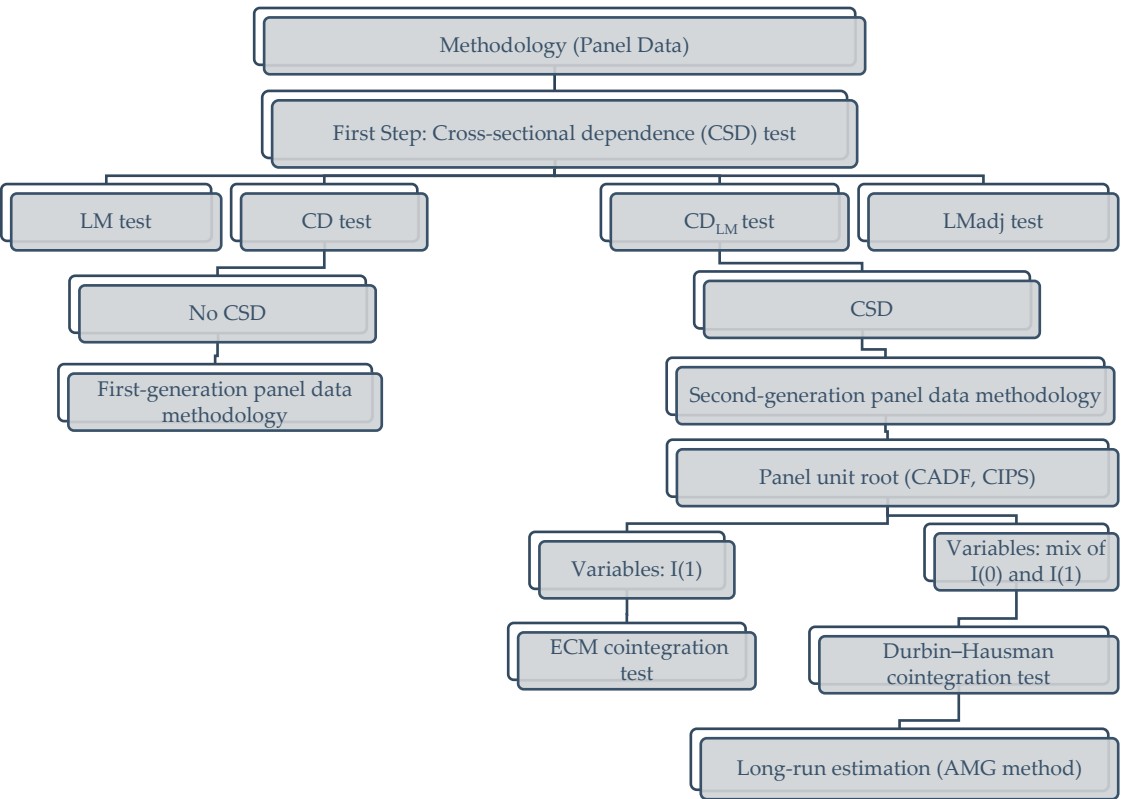

**Figure 2.** Summary of the methodological approach applied in the study.

## 5. Empirical Findings

In this section, the results are reported by following the econometric procedures mentioned above. Firstly, the presence of CSD between series was analyzed with four different tests. Table 2 presents the findings from the CSD tests. The null hypothesis of no CSD was rejected in all applied tests. As seen from the results, the null hypothesis was rejected, indicating the presence of CSD among BRICS-T countries, and that a shock occurring in one of these countries may affect other countries as well.

After the CSD examination, the unit root levels of the variables should be determined. As the previous results indicated CSD, applying a unit root test that takes this situation into account will provide reliable results. The study utilized the Pesaran [69] CADF test to analyze the unit root levels of variables, with consideration of CSD. Table 3 reflects the results, indicating that GINI, FP, EG, and T are stationary at the first difference level, whereas IQ and GLB are stationary at the level. Hence, GINI, FP, EG, and T exhibit I(1) characteristics, while IQ and GLB exhibit I(0) characteristics. The CADF findings suggest that using fixed effects and random effects as first-generation panel data methods may result in a spurious regression issue, as highlighted by Bulut et al. [73]. Therefore, it is

crucial to examine whether there is a cointegration relationship among variables in order to eliminate this issue.

**Table 2.** CDS results.

| Variable | CSD Tests | | | |
| --- | --- | --- | --- | --- |
| | **LM** | **CD$_{LM}$** | **LM$_{adj}$** | **CD** |
| lnGINI | 150.28 * | 24.70 * | 24.56 * | 12.48 * |
| lnIQ | 57.59 * | 7.77 * | 7.63 * | 5.41 * |
| lnFP | 77.73 * | 11.45 * | 11.31 * | 1.91 *** |
| lnGLB | 315.89 * | 54.93 * | 54.79 * | 17.76 * |
| lnEG | 318.33 * | 55.38 * | 55.24 * | 17.83 * |
| InT | 119.18 * | 19.02 * | 18.88 * | 5.54 * |

*, and *** indicate statistical significance at the 1%, and 10%, respectively.

**Table 3.** CADF results.

| | **t-Bar** | | **Z [t-Bar]** | | *p*-**Value** | |
| --- | --- | --- | --- | --- | --- | --- |
| | **Level** | **1st Difference** | **Level** | **1st Difference** | **Level** | **1st Difference** |
| lnGINI | −1.328 | −2.653 | 1.092 | −2.255 | 0.863 | 0.012 ** |
| lnIQ | −2.321 | - | −1.417 | - | 0.078 * | - |
| lnFP | −1.638 | −2.576 | 0.309 | −2.060 | 0.621 | 0.020 ** |
| lnGLB | −2.351 | - | −1.492 | - | 0.068 * | - |
| lnEG | −2.057 | −3.111 | −0.751 | −3.413 | 0.226 | 0.000 *** |
| InT | −2.185 | −3.497 | −1.072 | −4.387 | 0.142 | 0.000 *** |

The values ***, **, and * correspond to the levels of statistical significance at 1%, 5%, and 10%, respectively.

The existence of CSD and the results obtained from the unit root test are suitable for the application of the DH cointegration test. Table 4 reports the DH results, which indicate that the null hypothesis of no cointegration is rejected. The test statistics exceed the critical values, leading to the rejection of the null hypothesis. Thus, a conclusion is reached that there is a cointegration relationship among the examined variables. Consequently, income distribution policies can be designed in BRICS-T countries through institutional quality, press freedom, and other control variables.

**Table 4.** DH results.

| | **Statistics** |
| --- | --- |
| DHg | 7.314 * |
| DHp | 12.689 * |

Critical value is 2.33 for 1% (*)

After identifying the co-integration relationship, long-term coefficient estimations can be made. The AMG method was utilized to examine the impact of institutional quality, press freedom, and other control variables on inequality within the context of this framework. Table 5 demonstrates the coefficient estimations for the entire panel. Based on the results of the study, institutional quality has a negative and significant effect on inequality, as expected. A 1% increase in institutional quality in BRICS-T countries reduces the Gini coefficient by 0.072%. This result indicates that well-functioning high-quality institutions will act as a catalyst in reducing income inequality. These findings align with prior research such as Josifidis et al. [25], Szczepaniak et al. [26], and Kouadio and Gakpa [27]. The coefficient for press freedom, another basic independent variable, is positive and significant at the 10% level. As mentioned in Freedom House [64] data, high values indicate restrictions on press freedom. Therefore, the results show that a 1% decrease in press freedom increases income inequality by 0.059%. This result suggests that restrictions on press freedom in BRICS-T countries will have negative consequences for income distribution dynamics.

**Table 5.** AMG results (entire panel, dependent variable: GINI).

| Variable | Coef. | Std. Err. | *p*-Values |
|---|---|---|---|
| lnIQ | −0.072 | 0.029 | 0.013 ** |
| lnFP | 0.059 | 0.045 | 0.093 * |
| lnGLOB | 0.056 | 0.076 | 0.460 |
| lnEG | −0.025 | 0.067 | 0.754 |
| lnT | −0.025 | 0.015 | 0.091 * |

** and * refer to statistical significance at the 5% and 10% levels, respectively.

When looking at the control variables included in the model, the coefficient for globalization is positive but statistically insignificant. In fact, there are many studies that find globalization has a negative effect on income distribution, especially in developing countries [5,76,77]. Therefore, while the results are similar in terms of the coefficient sign as these studies, the lack of statistical significance means that a clear relationship between globalization and inequality cannot be established in these countries. Similarly, the coefficient for GDP per capita, which symbolizes economic growth, is negative as expected, but statistically insignificant. Milanovic [37], who has great prestige in inequality studies, states that economic growth is one of the most critical factors in reducing inequality. While the coefficient sign is consistent with this situation, the lack of statistical significance indicates that the fruits of economic growth are not distributed sufficiently in these countries. Finally, the coefficient for trade openness is negative and significant at the 10% level. A 1% increase in trade openness reduces the Gini coefficient by 0.025%. This result is partially consistent with Batuo et al. [16]. In these countries, trade openness may reduce inequality by creating more demand for low-skilled labor and improving wages.

Table 6 reports the long-term coefficient estimates obtained through the AMG for each country. It can be seen that the results obtained are quite heterogeneous and vary from country to country. The results show that institutional quality has a negative and significant impact on inequality in Brazil, India, and South Africa. Improving institutional quality in these countries can play a key role in ensuring fairness in income distribution. On the other hand, although the coefficient signs are consistent with expectations in Russia, China, and Turkey, statistical significance could not be achieved. Significant relationships were obtained for the institutional quality variable in three countries, while the significance of press freedom was achieved in four countries. Deterioration in press freedom in Brazil, Russia, South Africa, and Turkey disrupts income distribution. To put it differently, the presence of press freedom in these nations has a beneficial impact on the dynamics of income distribution by creating democratic channels.

**Table 6.** Country-level results (AMG, dependent variable: GINI).

| Countries | lnIQ | lnFP | lnGLB | lnEG | lnT |
|---|---|---|---|---|---|
| Brazil | −0.103 [b] | 0.064 [c] | 0.035 | −0.309 [a] | −0.050 [a] |
| | (0.010) | (0.080) | (0.639) | (0.000) | (0.000) |
| Russia | −0.067 | 0.122 [b] | 0.041 | −0.076 [a] | −0.091 [a] |
| | (0.187) | (0.015) | (0.695) | (0.000) | (0.000) |
| India | −0.198 [c] | −0.009 | 0.054 | 0.068 | 0.009 |
| | (0.060) | (0.826) | (0.727) | (0.562) | (0.822) |
| China | −0.038 | 0.248 | 0.389 [a] | 0.053 [c] | 0.002 |
| | (0.535) | (0.145) | (0.001) | (0.091) | (0.942) |
| South Africa | −0.020 [b] | 0.012 [c] | 0.007 | 0.169 [a] | −0.009 |
| | (0.048) | (0.096) | (0.615) | (0.000) | (0.260) |
| Turkey | −0.004 | 0.043 [b] | −0.189 [a] | −0.032 [c] | −0.009 |
| | (0.623) | (0.020) | (0.000) | (0.096) | (0.516) |

Statistical significance: 1% ([a]), 5% ([b]), 10% ([c]). Values in parentheses are *p*-values. The impact of each indicator constituting institutional quality on GINI is shown in Table A3 in Appendix C.

Similar to the findings obtained from the panel, the globalization variable is insignificant in 4 out of 6 countries. Statistical significance is only achieved in China and Turkey. This result indicates that the gains in China's globalization process have not spread to the general population, while in Turkey, there has been a dynamic that reduces inequality. Economic growth is statistically significant in all countries except India, but the coefficient signs show heterogeneity. Economic growth has been a factor that reduces inequality in Brazil, Russia, and Turkey. The results indicate that economic growth in these countries has spread to the general population and positively affected income distribution. Contrary to expectations, the results show that economic growth in China and South Africa does not have a completely egalitarian character. Finally, trade openness is only significant in Brazil and Russia. Coefficient estimates indicate that trade openness reduces inequality in both countries.

## 6. Discussions

In this section, the research findings will be discussed in detail. BRICS and Turkey have achieved higher economic growth than many developed countries since the 2000s. Especially in BRICS countries, the acceleration of investments in technological innovation has been the driving force behind high economic growth [78]. On the other hand, in Turkey, which has a similar characteristic with this group of countries, economic growth gained great momentum in this period with economic transformation and strong foreign capital inflows [79]. Despite the magnificent performance in economic growth, similar performance has not been achieved in socio-economic indicators in BRICS-T countries.

When looking at the entire panel, it was concluded that the institutional quality index reduced the Gini coefficient. In fact, this result is consistent with a significant part of the literature and is reasonable. Strengthening property rights through legal reforms enables these countries to attract more foreign investments. Foreign capital inflows and the reduction of political uncertainty rapidly increase investment and employment. As Stiglitz [12] expressed, the decrease in unemployment also leads to a fair income distribution. Similarly, strict enforcement of corruption control is an important parameter in reducing inequalities. ICRG data indicate that corruption is more widespread in these countries than in developed countries. In this sense, restricting the corrupt behavior of politicians and bureaucrats reduces inequality of opportunity and prevents distortions in resource allocation. Corruption control also reduces the power of elites. For example, some economic activities of elites can be exempt from taxes with the help of corrupt bureaucrats and politicians. Improving the quality of institutions in these countries can prevent a small minority from obtaining such exemptions and can create a more just tax system. In addition, increasing tax revenues can create resources for more effective redistribution policies [80].

On the other hand, improvements in institutional categories such as government stability, democratization, accountability, bureaucratic quality, and civil liberties reduce the influence of elite lobbying and enable the formation of more social policies. In this framework, a democratic, transparent, and stable government creates policies that prioritize social issues. If inequality is a significant issue, it designs more effective distribution and poverty alleviation policies. Therefore, well-functioning institutions can significantly affect economic and social inequality through all of these channels. Institutions spread power to society and can rein in income inequality by implementing effective policies.

It is possible to say that promising results have been obtained for countries as well. The study findings demonstrate that institutional structure plays a substantial role in reducing inequality in Brazil, India, and South Africa. Therefore, we can say that in these countries the institutional framework allows enjoying the socioeconomic benefits associated with institutional quality. Some well-functioning institutional factors in these countries manage to reduce market failures and reduce inequalities by maintaining redistributive policies more efficiently. On the other hand, although the coefficient signs in Russia, China, and Turkey are as expected, the results are insignificant. In these countries, institutional quality appears to be below the optimal threshold. Therefore, the effect of institutions is

not at a level to reduce income inequality. Within the framework of these results, further strengthening of institutions in these countries is very urgent.

Freedom of the press is positive and significant for the entire panel. This result indicates that a decrease in press freedom would increase inequality. In other words, expanding press freedom is functional in reducing inequality. Ensuring freedom of the press is the cornerstone of freedom of speech and democratic practices. Additionally, corruption control, political participation, and accountability are more easily achieved in the presence of free and strong media. The press exposes activities that are not in the public interest and ensures their elimination, thus serving the public interest. In this sense, it can prevent the exploitation of the poor by the elites, both economically and politically. Therefore, it distributes asymmetric power in society and contributes to spreading power to the masses. Shaping redistribution policies is not independent of power relations. The press can play a crucial role in reversing the increasing inequality resulting from inadequate redistribution policies by shaping public opinion and exerting pressure on the government. In other words, press organizations that address the problem of inequality in all its reality, criticize it, and reflect it to the public can play a critical role in reducing inequality through policy reforms.

The country-specific findings obtained indicate that the coefficient of press freedom index is positive and statistically significant in Brazil, Russia, South Africa, and Turkey. These results suggest that press freedom can have direct and indirect effects on inequality in these countries. In short, it can be said that the press's more proactive stance and continued information flow facilitate the implementation of policies that reduce inequality in these countries. In this framework, the press is successful in shaping public opinion. Society can put pressure on the government for economic issues. Public pressure, shaped by the press, motivates governments to fix issues. All of these developments indicate that the press has a role in reducing inequality in these countries. On the other hand, there is no significant effect of press freedom on inequality in China and India. In particular, it is a fact that the Chinese government puts great pressure on dissidents, activists, and journalists. An article published in Foreign Policy in 2016 mentioned that activities that expand press freedom in China are harshly punished [81]. A news article in The Economist in 2022 highlighted that there is an attempt at media monopolization in India and ultra-rich businesspeople are beginning to monopolize the media, reducing press freedom [82]. While the press should expose the privileges of the elite and work for the benefit of the public, monopolization by a wealthy minority will eliminate the press's function in reducing inequalities. In this context, the press is not free in an environment where journalists are threatened, and imprisoned and the media is monopolized. The liberalization of press activities in these countries should become one of the priority agenda items for achieving significant socioeconomic gains.

When looking at the effects of control variables in BRICS-T countries, heterogeneous results were obtained for both the entire panel and individual countries. Globalization is not significant in the entire panel solution. Moreover, it is statistically insignificant in countries other than China and Turkey. In fact, these countries have been an important part of the economic globalization process since the late 1990s. In this context, it is expected that globalization would have a positive or negative effect on inequality in these countries, but the results obtained are quite surprising. On the other hand, the effect of globalization on inequality in Turkey is negative and significant. In other words, Turkey's integration into the globalization process has been one of the factors that reduce income inequality. On the other hand, the results show that China's globalization process benefits mostly the upper classes and increases income inequality.

Although economic growth is statistically insignificant for the entire panel, it is significant in all countries except India. The coefficient is negative in Brazil, Russia, and Turkey. This result can be interpreted as the economic growth process reducing inequality by creating employment opportunities and implementing more effective redistribution policies. In other words, the returns of economic growth in these countries can trickle down. On the

other hand, the coefficients are positive in China and South Africa. Jain-Chandra et al. [83] noted that China's growth process has significantly reduced poverty in the last twenty years but has not been successful enough in reducing inequality. The authors particularly pointed out that urbanization, which has increased since the 1980s, has been a major factor in the increase in income inequality. Although there has been a partial decrease in inequality indicators since 2010, justice in income distribution cannot be achieved due to the lack of strong structural reforms. Similarly, the fruits of growth in South Africa have not been distributed evenly across society. This result indicates that growth alone cannot reduce inequality in South Africa. The negative and significant coefficient of institutional quality for South Africa can be interpreted as growth being able to distribute more equally with strong reforms and well-functioning institutions.

Finally, trade openness is statistically significant and negative for the entire panel. However, in country-specific estimations, it is only statistically significant in Brazil and Russia, with negative coefficients in both countries. Anyanwu et al. [9] indicate that increasing trade openness can increase the demand for low-skilled labor in developing countries, resulting in a significant improvement in wages. On the other hand, it is stated that due to the decrease in the demand for high-skilled labor, wages in this group can decrease and therefore income distribution can improve. In fact, both Brazil and Russia are integrated into international trade with energy and agricultural products. It is plausible that trade openness without high technology can improve income distribution by improving the wages of low-skilled labor in these countries. On the other hand, trade openness has an insignificant effect on inequality in China, India, South Africa, and Turkey. The results are consistent with Dabla-Norris et al. [18] and Milanovic [14]. This result can be interpreted as the trade composition (traditional or technological) in these countries does not significantly affect inequality.

As known, data constraints have caused the analysis to be completed in the year 2016. Therefore, for the post-2016 period, up-to-date discussion and inference is required. To make some inferences related to the research question in the BRICS-T countries considered after 2016, it might be useful to look at some recently published data. Looking at the recently dated Gini coefficients published by SWIID, it is observed that the income distribution dynamics in BRICS-T countries have not changed significantly. The current levels of the Gini coefficient are almost the same as in 2016. In fact, it is quite probable and normal not to have a significant change in the Gini coefficient in a short period of 5–6 years. Even though there are no recent data on both press freedom and institutional quality, it might be possible to make inferences about the current situation of the countries discussed from the democracy index announced by Freedom House [84] for each country. In this sense, according to Freedom House [84], only Brazil and South Africa have free status (F) among the 6 countries considered. On the other hand, while India is partly free status (PF), Russia, China, and Turkey are not free status (NF). In line with the results, it points out that expanding freedoms can be an important factor in reducing socioeconomic problems, given the current status of countries. That is, of the three countries where institutional quality reduces income inequality, two are free and one is partially free. On the other hand, in non-free countries, the ability of institutional quality to solve social problems such as income inequality is weak. Likewise, the results for freedom of the press are quite similar. In other words, issues such as institutional quality, freedoms, democratic practices, and social justice do not seem to be independent of each other. On the contrary, it can be clearly said that there are processes that support each other. In this sense, although the BRICS-T countries have not made any substantial progress in the last few years in terms of institutions, democracy, and equality, the findings of the study should be considered very important in terms of implying that developments in institutions and freedoms can be an antidote to economic/social equality. This is indeed an opportunity for policymakers in BRICS-T countries. Because the promotion of these countries to the status of developed countries cannot be achieved only with high economic performance. Economic growth should be crowned with achievements in areas such as institutional quality, democracy, and social

justice. At this point, the results show that inequality can be reduced through improvements in institutional quality and freedom of the press. This is actually an important win-win opportunity and should be used by these countries.

## 7. Conclusions and Policy Recommendations

This study is the first attempt to examine the impact of institutional quality and press freedom on income inequality. With this aim in mind, the potential effects of institutional quality and press freedom on inequality were explored first. After discovering the potential mechanisms, the long-term effects of institutional quality, press freedom, globalization, economic growth, and trade openness on inequality were examined using the AMG method for the BRICS-T countries during the period 1993–2016. The results obtained for the entire panel indicate that institutional quality has a negative and significant effect on inequality. At the same time, the findings show that expanding press freedom is a factor that reduces inequality. When looking at the control variables, it can be seen that globalization and economic growth are not statistically significant for the entire panel. On the other hand, trade openness has a negative and significant effect on inequality.

The findings obtained for each country are not homogeneous, but they indicate that institutional quality and press freedom can have significant effects on inequality. Institutional quality is seen as a factor that reduces inequality in Brazil, India, and South Africa. Although the coefficient of institutional quality is negative in other countries, there is no statistical significance. Moreover, press freedom appears as a factor that reduces inequality in four out of the six countries examined. Developments in press freedom in Brazil, Russia, South Africa, and Turkey have a positive impact on income distribution dynamics. When looking at control variables, globalization increases inequality in China but reduces it in Turkey. On the other hand, economic growth is statistically significant in all countries except India, but the coefficient signs are quite heterogeneous. The results indicate that economic growth leak down in Brazil, Russia, and Turkey, while growth and inequality act simultaneously in China and South Africa. Finally, trade openness appears to be a factor that softens inequality in Brazil and Russia.

In fact, the estimates made for both the panel and the countries are quite promising. BRICS-T countries' attainment of developed country status cannot be achieved only with high economic growth. Findings from the study show that these countries have significant opportunities to reach developed country status. The most important elements that characterize developed countries are well-functioning institutions, strong democracy, and an egalitarian structure. Evidence indicates that BRICS-T countries can curb inequality by developing the institutional structure and freedom of the press. Thus, policymakers have the opportunity to reduce inequality while improving institutional quality and freedom of the press.

Within the framework of all these realizations, policy recommendations can be made for BRICS-T countries. When the institutional quality is low, the policy design skills of the institutions are weak. With this weakness, income inequality cannot be combated. Policymakers should build institutions with sound and decisive reforms to reduce inequality. In this context, first of all, the legal system needs to be improved. Guaranteeing property rights and contracts is essential to the creation of economic gains. Stronger property rights motivate economic growth by increasing national and international investment. It also facilitates a sustainable trickle down of economic growth. In addition, preventing corruption and increasing the bureaucratic quality allow the reduction of harmful activities for the society and the protection of the benefit of the society. Thus, the exploitation of the poor by powerful groups can be prevented. In addition, the improvement in these categories and government efficiency will ensure that the problems in society are correctly identified and efficient policies are formed. Especially Russia, China, and Turkey should implement more stable and robust regulations for the development of the institutions' structure. In summary, the evidence suggests that sustaining institutional quality in BRICS-T is expected to yield many positive benefits as well as a fair distribution of income. For this reason, the

public, non-governmental organizations, and bureaucrats should never compromise on increasing institutional quality.

Finally, it is necessary to consolidate freedoms and democracy in all areas, including the press. In this framework, policymakers should guarantee civil rights and freedom of expression in all spheres, both public and non-public. Institutions examining the press should be independent institutions that are not under the influence of political power. Otherwise, it may be easier for the government to put pressure on the press. At the same time, preventing monopolization in the media sector will foster freedom of the press and polyphony. Ensuring freedom of the press will be a catalyst for the elimination of power asymmetry and the spread of power to the public. Through these policies, the enhancement of press freedom facilitates the political organization of disadvantaged/losing groups. This organization puts pressure on policymakers and more efficient redistribution policies are designed.

Despite the spectacular economic performance of these countries, the fact that income inequality has not been reduced significantly is a remarkable situation for policymakers. This situation should be resolved with mechanisms that ensure the spread of economic gains to the whole society. In this framework, policymakers can distribute the fruits of growth more equitably through progressive taxation and strong social transfers. In addition, although there has been an increase in access to education in BRICS-T in recent years, increasing the quality of education can improve the qualifications of people and enable them to earn more income. In other words, policymakers can improve the education system and ensure that the market mechanism creates a more equitable distribution.

This study is the first attempt to examine the effect of freedom of the press on income distribution. In this respect, it can lead to a new research network in the literature. It is important to acknowledge that the study has certain limitations. Firstly, due to the lack of current data, the analysis period has been determined as 1993–2016. The lack of current data can make it difficult to make a healthy inference, especially in Russia. Some significant changes in the world economy, particularly over the last 10 years, could not be interpreted due to the absence of current data. Although this study focuses on the BRICS-T countries as a whole, there are some important issues regarding Russia, an important member of the group. The Russia–Ukraine War puts the military needs of this country at the forefront. The war conditions militarizing society could weaken the power of the press to positively influence social policies. Therefore, future studies should expand the analysis period and reveal the effects of such changes in the world economy more clearly with the publication of current data. In addition, future studies may do more to clarify the potential mechanisms between inequality, institutions, and freedom of the press. In addition, obtaining fresh empirical evidence for different countries and groups of countries can enrich the literature. Finally, it can be exciting to theoretically and empirically examine the influence of institutions and press freedom on wealth inequality.

**Funding:** This research did not receive any specific grant from funding agencies in the public, commercial, or not-for-profit sectors.

**Institutional Review Board Statement:** Not applicable.

**Informed Consent Statement:** Not applicable.

**Data Availability Statement:** The data sources are publicly available and accessible through these sources.

**Acknowledgments:** I express my gratitude to Reviewer 1 who drew attention to the point that the military needs caused by the war atmosphere in Russia might weaken the power of the press to influence social policies. I would also like to thank the editor and 4 reviewers for their valuable comments and suggestions.

**Conflicts of Interest:** I declare that there are no conflict of interest.

## Appendix A

As a contribution to Figure 1, the potential mechanisms/functions and tools of press freedom on income equality are provided in Table A1. The left side of the table expresses the mechanism of press freedom on income distribution, while the right side explains how this influences income distribution through various tools.

**Table A1.** Potential mechanisms/functions and tools of press freedom on income distribution.

| Mechanisms/Functions | Tools |
|---|---|
| Public interest | It exposes activities that are not in the public interest, ensures the cessation of such activities through public pressure, and thus upholds the public interest. Thus, actions/policies that would disrupt income distribution are eliminated, prioritizing the fair distribution of income. |
| Fair distribution of power | It prevents the concentration of political and economic power in certain sections, disperses power to the society, and prevents the implementation of income distribution policies that are against the poor (for example, non-implementation of wealth tax, proliferation of indirect taxes). |
| Accountability/transparency | By disseminating true information to society, it enables the public to demand transparency and accountability, making policymakers transparent and accountable, thus contributing to the transparent implementation of redistribution policies. |
| Efficient policy-making | It establishes an informal control mechanism over institutions/bureaucrats, ensures corruption control, increases bureaucratic quality, and thus enables the implementation of more efficient policies including policies aimed at improving income distribution. |

## Appendix B

Although information about the variables is provided in the model and data definitions sections, it is believed that Table A2 would be useful to give detailed information about the calculation method of the indicators. Additionally, the recommended sources can be reviewed for detailed information.

**Table A2.** Detailed explanation of the variables and indicators.

| Variables | Indicators/Indexes | Calculation Method | Data Sources |
|---|---|---|---|
| GINI | Gini coefficient index | Inequality in disposable (post-tax, post-transfer) income | SWIID; For detailed information, see: Solt [62] |
| IQ | Government stability | The allocated risk rating is the aggregate of three different elements (government unity, legislative strength, popular support), each scoring between 0 and 4. Four points indicate an extremely low risk, while zero points suggest an extremely high risk. | ICRG; For detailed information, see: ICRG methodology |
| | Bureaucracy quality | Countries where the bureaucracy has the strength and expertise to govern without drastic interruptions in state services are awarded high scores. Otherwise, the country is scored low. It takes a value between 0 and 4. | ICRG; For detailed information, see: ICRG methodology |
| | Corruption control | This evaluation pertains to the level of corruption within the political structure. This corruption endangers foreign investment due to a number of factors. It varies between 0 and 6. 6 indicating high corruption control. | ICRG; For detailed information, see: ICRG methodology |
| | Military in politics | Its participation in politics, even if marginal, represents a reduction in democratic responsibility. In essence, lower risk scores point to increased military interference in political matters, leading to elevated political risks. It takes values between 0 and 6. | ICRG; For detailed information, see: ICRG methodology |

**Table A2.** *Cont.*

| Variables | Indicators/Indexes | Calculation Method | Data Sources |
|---|---|---|---|
| | Democratic accountability | This is a measure of how responsive government is to its people. It varies between 0 and 6. Higher values are higher democratic accountability. | ICRG; For detailed information, see: ICRG methodology |
| | Law and order | 'Law and order' stand as one unit; however, its dual aspects ('law' and 'order') undergo separate evaluations, with each facet receiving a score ranging from 0 to 3 points. Superior scores reflect a more efficient judicial system | ICRG; For detailed information, see: ICRG methodology |
| FP | Press freedom index | Press freedom in every nation and region is assessed through 23 methodological questions split into three main areas: legal, political, and economic environments. The final score for a country or territory (ranging from 0 to 100) is the cumulative sum of the points assigned for each question. | Freedom House; For detailed information, see: Freedom House [64] |
| GLB | Globalization index | The Globalization Index is a comprehensive measure that gauges globalization across all countries, taking into account economic, social, and political aspects. The Index is grounded on 43 variables. | KOF Swiss Economic Institute; For detailed information, see: Gygli et al. [65] |
| EG | Economic growth | GDP per capita (constant 2015 USD) | WDI |
| T | Trade openness | Proportion of total exports and imports to GDP | WDI |

**Appendix C**

Table A3 shows the impact of all indicators on income inequality for each country. Unlike Table 6, it points to the impact of each indicator constituting institutional quality on income inequality. In this framework, it helps to analyze the main institutional indicators behind income inequality. In fact, the results in Table A3 are quite consistent with Table 6. In Brazil, government stability, corruption control, democratic accountability, and law/order appear to be very effective in reducing income inequality. On the other hand, press freedom (a decrease in the index) is also a factor in reducing income inequality. In Russia, the main institutional mechanisms that help to reduce income inequality are military in politics and law/order. Similar to Table 6, press freedom reduces inequality. While globalization is statistically significant, trade openness is insignificant compared to Table 6. In India, government stability, corruption control, and law/order are institutional quality indicators that reduce income inequality. Press freedom is insignificant. China distributes income more equitably through corruption control and democratic accountability. Press freedom appears to be dysfunctional. Unlike Table 6, trade openness is statistically significant. In South Africa, all indicators except bureaucratic quality and military in politics reduce income inequality. Press freedom is also positive and significant here. In other words, the press is a catalyst that reduces income inequality. Finally, in Turkey, corruption control and law/order reduce income inequality. Otherwise, press freedom serves a function of reducing inequality.

The results indicate that the indicators of law/order and corruption control are particularly fundamental drivers in reducing income inequality. Democratic accountability and government stability also appear important. On the other hand, bureaucratic quality has not succeeded in reducing income inequality in any country. Lastly, press freedom is an actor in reducing income inequality in Brazil, Russia, South Africa, and Turkey.

**Table A3.** General conclusions of the analysis of the press institute in overcoming income inequality for the period 1993–2016.

| Variables | Indicators/Indexes | BRICS-T Countries | | | | | |
|---|---|---|---|---|---|---|---|
| | | **Brazil** | **Russia** | **India** | **China** | **South Africa** | **Turkey** |
| lnIQ | Government stability | **−0.047 **** (0.027) | 0.009 (0.716) | **−0.082 **** (0.014) | −0.019 (0.606) | **−0.026 **** (0.042) | −0.011 (0.582) |
| | Bureaucracy quality | −0.055 (0.292) | −0.045 (0.290) | −0.024 (0.364) | −0.024 (0.268) | 0.067 (0.259) | 0.009 (0.801) |
| | Corruption control | **−0.038 ***** (0.084) | −0.013 (0.454) | **−0.026 **** (0.018) | **−0.248 **** (0.014) | **−0.022 *** (0.009) | **−0.020 ***** (0.099) |
| | Military in politics | 0.018 (0.854) | **−0.103 *** (0.001) | 0.036 (0.495) | 0.013 (0.199) | 0.026 (0.191) | 0.012 (0.125) |
| | Democratic accountability | **−0.008 *** (0.004) | 0.011 (0.548) | −0.007 (0.188) | **−0.062 *** (0.000) | **−0.037 ***** (0.096) | −0.004 (0.663) |
| | Law and order | **−0.063 ***** (0.057) | **−0.058 ***** (0.089) | **−0.130 *** (0.001) | 0.013 (0.777) | **−0.107 *** (0.001) | **−0.036 **** (0.045) |
| lnFP | Press freedom index | **0.151 **** (0.045) | **0.235 *** (0.000) | −0.022 (0.785) | 0.042 (0.713) | **0.182 ***** (0.059) | **0.116 *** (0.000) |
| lnGLB | The Globalization Index | 0.056 (0.710) | **−0.670 **** (0.022) | 0.033 (0.914) | **0.800 *** (0.000) | 0.125 (0.248) | −0.091 (0.413) |
| lnEG | GDP per capita (constant 2015 US$) | **−0.353 **** (0.014) | **−0.045 **** (0.012) | 0.036 (0.664) | **0.071 **** (0.026) | **0.148 ***** (0.094) | **−0.059 **** (0.020) |
| lnT | Proportion of total exports and imports to GDP | **−0.080 *** (0.007) | −0.006 (0.856) | 0.046 (0.544) | **0.063 **** (0.020) | −0.008 (0.985) | 0.0017 (0.994) |

Parentheses show the *p*-value. *, **, and *** indicate statistical significance at the 1%, 5%, and 10% level, respectively.

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
