# Peer review of "Income Inequality, Institutions, and Freedom of the Press: Potential Mechanisms and Evidence"

_sustainability, doi:10.3390/su151712927_

Round 1

Reviewer 1 Report

The article contains a number of comments and recommendations that must be fully taken into account. They relate to the methodology, the research period, taking into account current trends in the world economy, and more.

Thank you!

Author Response

The revision report is attached as a word file.

Reviewer 2 Report

Interesting article. Well presented research problem. Appropriate conclusions. Sufficient literature discussion. This section can be improved.

Author Response

(The authors gave the same response as above.)

Reviewer 3 Report

I would recommend to change 'Prob.' in Table 5 and 'probabilities' in Table 6 (footnote) to 'p-value(s)".

Change 'a', 'b' and 'c' in Tables 3 and 5 to '***', '**' and '*'.

Author Response

(The authors gave the same response as above.)

Reviewer 4 Report

I am sympathetic with this paper, showing a nice and almost original idea.

I would see a clarification of the methodology adopted in the paper, which is sometimes implicit in the writing, and a better justification of the rationale behind.

Literature review is relatively wide but can be enlarged a bit more, you have space to do so.

Not sure the conclusions have a sufficient depth for policy analysis and implementation, so I would encourage authors to enlarge this part.

Language usage needs some adjustments in specific parts of the manuscript, you can argue that especially when sentences are very long, and the readers can misunderstand the main concept at stake.

Some sparse revisions needed

Author Response

(The authors gave the same response as above.)

Round 2

Reviewer 1 Report

The provided recommendations are taken into account in the review.

I do not recommend working on scientific problems without taking into account current changes, because the future is more important than the past.

Thank you!

Reviewer 4 Report

Standard revisions, thank you for your documented effort.

Some revisions still needed.